# Achieving the Rewards of Smart Agriculture

**Jian Zhang** [1,2,*,†] [iD], **Dawn Trautman** [3,†], **Yingnan Liu** [4], **Chunguang Bi** [1,5], **Wei Chen** [4], **Lijun Ou** [6,*]
**and Randy Goebel** [7,*] [iD]

1 Faculty of Agronomy, Jilin Agricultural University, Changchun 130018, China; chunguangb@jlau.edu.cn
2 Department of Biology, University of British Columbia, Kelowna, BC V5K1K5, Canada
3 Nuffield Canada, Edmonton, AB T5K 1X5, Canada; det@ualberta.ca
4 Heilongjiang Academy of Science, Harbin 150040, China; liuyingn234@163.com (Y.L.);
   applechenwei@163.com (W.C.)
5 College of Information Technology, Jilin Agricultural University, Changchun 130018, China
6 College of Horticulture, Hunan Agricultial University, Changsha 410128, China
7 Department of Computing Science, Alberta Machine Intelligence Institute, University of Alberta,
   Edmonton, AB T6G 2R3, Canada
* Correspondence: jian.zhang@ubc.ca (J.Z.); ou9572@hunau.edu.cn (L.O.); rgoebel@ualberta.ca (R.G.)
† These authors contributed equally to this work.

**Abstract:** From connected sensors in soils, on animals or crops, and on drones, to various software and services that are available, "smart" technologies are changing the way farming is carried out. These technologies allow producers to look beyond what the eye can see by collecting non-traditional data and then using analytics tools to improve both food sustainability and profitability. "Smart Agriculture/farming" (SA) or "Digital Agriculture" (DA), often used interchangeably, refer to precision agriculture that is thus connected in a network of sensing and acting. It is a concept that employs modern information technologies, precision climate information, and crop/livestock developmental information to connect production variables to increase the quantity and quality of agricultural and food products. This is achieved by measuring and analyzing variables accurately, feeding the information into the cloud from edge devices, extracting trends from the various data, and subsequently providing information back to the producer in a timely manner. Smart agriculture covers many disciplines, including biology, mechanical engineering, automation, machine learning, artificial intelligence, and information technology-digital platforms. Minimum standards have been proposed for stakeholders with the aim to move toward this highly anticipated and ever-changing revolution. These foundational standards encompass the following general categories, including precise articulation of objectives, and baseline standards for the Internet of Things (IoT), including network infrastructure (e.g., stable 4G or 5G networks or a wireless local area network (WLAN) are available to end users). To sum up, SA aims to improve production efficiency, enhance the quality and quantity of agricultural products, reduce costs, and improve the environmental footprint of the industry. SA's ecosystem should be industry self-governed and collaboratively financed. SA stakeholders and end-users' facilities should meet standard equipment requirements, such as sensor accuracy, end data collectors, relevant industry compliant software, and trusted data analytics. The SA user is willing to be part of the SA ecosystem. This short perspective aims to summarize digital/smart agriculture concept in plain language.

**Keywords:** digital agriculture; artificial intelligence; internet of things (IoT); machine learning; image processing and analysis; genomics and phenomics

## 1. Introduction

With the population expected to reach 9.7 billion by 2050, the world is facing many challenges, sustainable food production chief among them. The Food and Agriculture Organization of the United Nations predicts that we need to boost worldwide food production by 70 percent over the next two decades to feed the anticipated population in

2050 [1]. Beyond population growth, there are additional drivers to produce more with less, and to produce food sustainably and responsibly while managing the impacts of climate change, consumer demographics, and preferences, as well as globalization and trade [2]. Growing consumer awareness is driving demand for high quality products, produced with sustainable practices, and with health-related attributes at the front of many consumers' minds (e.g., fewer herbicides, pesticides, antibiotics, implants administered, and with more regulatory oversight), not to mention consumer demand for traceability and attribute assurance.

Climate initiatives to reduce food waste and the consumption of high-resource products are becoming more prevalent in both developed and developing countries. The growing global population is also becoming more urban, and as more people move to cities, they also move into the middle class so demand for protein foods increases. Furthermore, cultural and generational influences are creating a fragmentation of tastes which marks an opportunity for market diversification, with some challenges for the traditional agri-food market [3]. In developed markets, growth drivers include health, nutrition, and sustainability. As incomes rise, "better for you foods" are replacing traditional "feel-good" junk foods. Related to this transition is a shift to plant-based foods, including plant-based meat, sometimes marketed as "clean meat".

All things considered, the need for innovation in agriculture has never been greater (cf., Figure 1). On the production side and across the value chain, businesses are increasingly pressured by rising input costs (e.g., seed, fertilizer, and labor), climate-driven changing land use priorities, and consumer demands for transparency and sustainability [4]. With the historically unpredictable and overall thin margins in agriculture, there is a need for new solutions, both for producers and consumers. As commodity prices continue to fluctuate, and in some cases, stagnate, there is intense awareness that new solutions are needed to provide relief, for both producers and consumers.

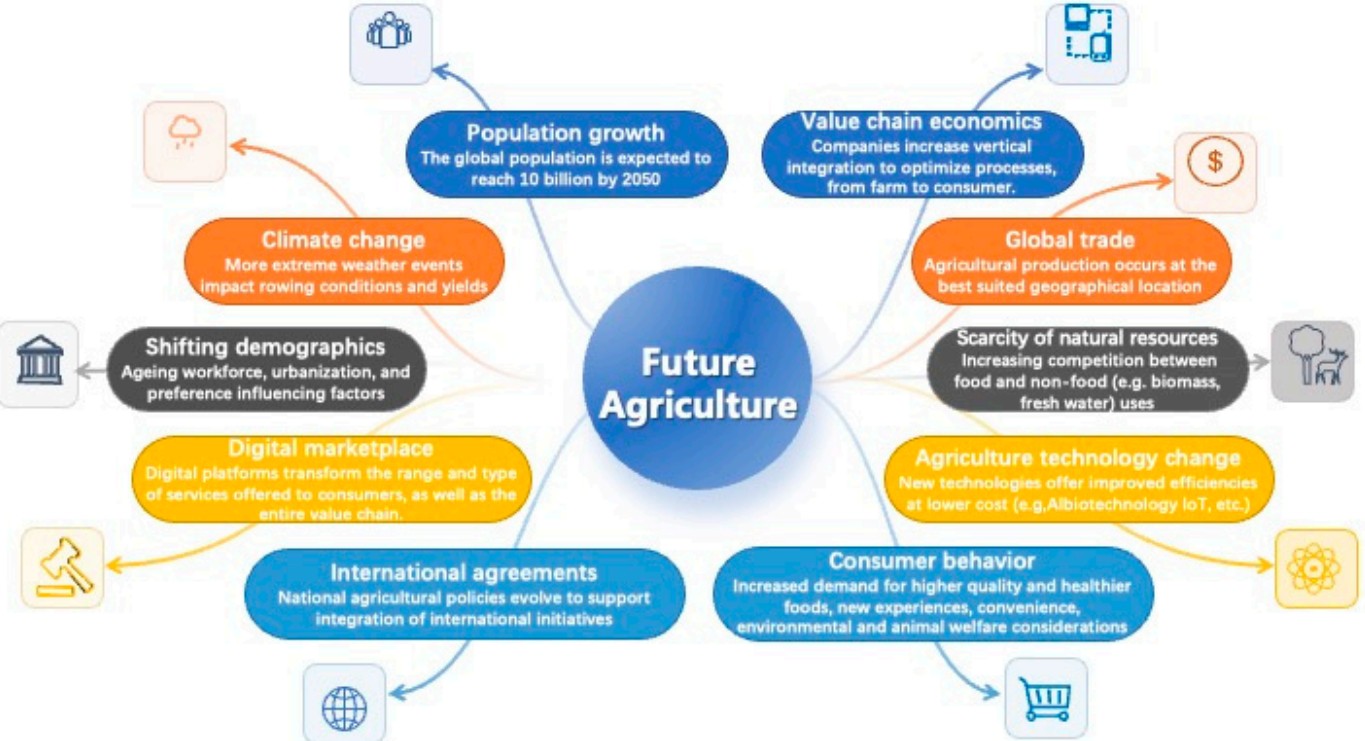

**Figure 1.** Ten global megatrends will lead to a disruptive transition in the next 5–10 years (Adapted from [5]).

One approach to producing more food with increasingly limited resources will be to improve efficiencies in resource use, including people, production technologies, and waste reduction in the agri-food supply chain. Note that this was largely the approach in the Green Revolution of the 1950s, with the adoption of systematic breeding, high-yielding varieties, synthetic fertilizers and pesticides, and best management practices. But over the past five to seven years, a new set of practices has emerged, which uses information and communication technologies and has created a digital revolution in agriculture; connected sensors, the Internet of Things (IoT) [6], autonomous vehicles, robots, and big data analytics [7] will be essential in effectively feeding tomorrow's world. The future of agriculture will be smart, connected, and digital. Understanding the challenges and opportunities in "Smart Agriculture" (SA) will help guide relevant policy decisions, agri-food businesses, and agri-food industry workers.

Smart agriculture incorporates current advanced technologies into existing farming or agricultural practices in order to increase production efficiency, improve the quality of agricultural products, reduce production costs, and reduce the environmental footprint. The distinction between "smart" agriculture and agriculture using digital technologies and precision management is based on the connectivity of data, using the Internet of Things (IoT) (see Section 3 for definitions and concepts in SA) [8]. Smart technologies in agriculture also have the capacity to improve the quality of life of farm workers by reducing heavy labor and tedious tasks (e.g., by using automation, robotics, and data analytics for precision management). Further, as labor can be one of the highest costs within production, and there being fewer agricultural laborers available, reducing our reliance on human labor will lead to further productivity gains.

Because of these developments, there is increased interest in and demand for proof of the economic value of digitization and connectivity in agriculture. Agriculture companies— from startups to original equipment manufacturers (OEMs)—are investing in the space. According to AgFunder, 2014 was the year in which AgTech truly launched with 2.36b USD being invested across 264 deals [9]; in AgFunder's most recent report, they show that 2018 saw investments in AgriTech of over 51b USD across 3155 deals. While there are numerous startups in the sector, some of the most notable investors include companies such as Google Ventures, Microsoft, and Bayer. So, in less than 10 years, improvements in the underlying technologies have led investors to consider the potential for SA. It is anticipated that the next few years will bring a consolidation of the sector with larger companies making investments, cost reductions across the life sciences, and improvements in sensor technologies, robotics, automation, and computation being deployed [10]. In summary, agriculture will become intelligent, interconnected, and digitized in the future. SA can reduce the dependence of agriculture on labor, improve resource utilization efficiency, alleviate the pressure on producers and consumers, and has unlimited potential in the future.

## 2. Definitions of Smart Agriculture

The modernization of agriculture and the adoption of various technologies in production practices have resulted in new concepts in agriculture, including precision agriculture, digital agriculture, and smart agriculture. While in some of the literature these terms are used interchangeably, there are differences in the details, which become important when defining "Smart Agriculture".

For example, the International Society of Precision Agriculture organization defines "Precision Agriculture as "... a management strategy that gathers, processes and analyzes temporal, spatial and individual data and combines it with other information to support management decisions according to estimated variability for improved resource use efficiency, productivity, quality, profitability and sustainability of agricultural production". In brief, SA is the application of connectivity (e.g., IoT) to traditional agriculture, where the use of sensors and software is used to collect data to optimize agricultural production decisions through mobile platforms, such that traditional agriculture becomes "smarter".

In a broad sense, smart agriculture also includes agricultural e-commerce, food traceability and anti-counterfeiting, agricultural leisure tourism, agricultural information services, and other aspects.

Precision agriculture, as important component of smart agriculture, is a method of farm management where the input needs of individual crops, fields, and livestock are optimized through data gathering, observation, and analysis. Big data and advanced analytics, as well as robotics, imagery, sensors, and weather forecasts enable precision agriculture to be an extremely effective agricultural management strategy [11]. For example, by assessing the conditions and needs of individual crops, fields, and livestock, precision agriculture can increase the quantity and quality of agricultural output while reducing inputs of water, energy, fertilizer, pesticides, supplements, etc., in order to save costs and reduce the environmental impact [12].

Smart agriculture (SA) entails the exploitation of data to optimize agricultural systems. The distinction from precision agriculture may seem minor, but is crucial, as it defines the next revolution within agricultural and digital industries. The focus of SA is on data exploitation; this requires access to data, data analysis, and the application of the results over multiple (ideally, all) farm or ranch operations. On-farm and on-ranch decisions are made using smart, mobile devices to access data on conditions, climate, input use, and labor, and more in real time, allowing the producer to subsequently make informed decisions based on validated data. SA implies a system of connectivity (e.g., Internet of Things, edge devices, low-power wide area network (LPWAN), 5G, etc.), where "things" are connected in a network of communication. The data (i.e., the information) are processed in farm management software (i.e., apps and online platforms) in a timely manner and relayed back to the producer to make informed decisions [13].

The International Organization for Standardization (ISO) defines SA as the "combination of network connectivity, widespread sensor placement, and sophisticated data analysis techniques [which] now enables 'smart farming' due to large amounts of data generated by IoT devices" [14]. For the purposes of this report and in an effort to standardize the concept of SA, the following definition will be used: SA uses Internet of Things (IoT) technology; various sensors are placed on equipment, livestock, or in the field to relay data to a platform (i.e., cloud-based; that is, using WLAN, edge devices, or to the cloud directly) allowing the creation of an information system for users (e.g., including farmers, agro-supply professionals, consultants, researchers).

## 3. Basic Concepts and Technologies

### 3.1. Related Terminology, Concepts, and Technology

Beyond the technical jargon of SA, there are additional related technologies and social concepts that support the idea of SA which are worth mentioning.

Agriculture 4.0 is a term used by the World Government Summit (2018) [15]; as an analogy to Industry 4.0 [16]. It refers to the integrated internal and external networking of farming operations as a result of the emergence of smart technology in agriculture [17]. The trend is towards automation, intelligence, and data exchange in agriculture. Agriculture 4.0 aims to disrupt the current food and agriculture regime and sustainably improve production capacity by using new techniques and technologies in production (e.g., hydroponics, algae feedstock, bioplastics, desert agriculture, seawater farming). The intent is to use novel technologies to improve food access, and change the current food supply chain (e.g., vertical/urban farming, genetic modification, 3D printing), and to incorporate cross-industry technologies and applications (e.g., UAVs, IoT, data analytics, PA, nanotechnology, blockchain, AI/ML, food sharing, and crowd farming) (World Government Summit, 2018) [5].

Based on the definition of Agriculture 4.0, the concept of "SA" generally resides within the third category: incorporating cross-industry technologies and applications; these inter-related technologies will be briefly defined in the following section [18].

Geographic Information Systems (GIS) are included within Agriculture 4.0, and are characterized by making significant use of global positioning system (GPS) technology. The latter allows plots to be mapped for increased precision during technical operations. For example, mapping the nitrogen requirements of a plot allows the inputs to be tailored not just at plot-level but for different zones within it identified by satellite. In addition, GIS systems can provide a framework for the deeper analysis of a broader scope of inputs, including weather patterns, topological features, and watershed models [19].

### 3.2. Big Data Analytics, Artificial Intelligence, and Machine Learning

In short, the application of Internet of Things technology to traditional agriculture, the use of sensors and software to control agricultural production through mobile platforms or computer platforms, so that traditional agriculture is more 'wisdom'. In addition to accurate perception, control, and decision management, in a broad sense, smart agriculture also includes agricultural e-commerce, food traceability anti-counterfeiting, agricultural leisure tourism, agricultural information services and other aspects.

The anticipated value of artificial intelligence and machine learning (AI) for Digital Agriculture is not different to that for general systems biology, materials science, drug design, or knowledge-based design in the transformation of genotype to phenotype.

Overall, the value of AI methods lies in those arising from the translation of theoretical advances to application tools that accelerate the creation of predictive models (e.g., with existing tools like Google's Tensorflow, Facebook's PyTorch). These application frameworks are designed to accelerate the creation of predictive models by transforming application data (e.g., labelled data instances like genomic samples from crop phenotypes).

It is important to note that the application of AI is possible in all areas of scientific research; overall, the idea is to capture and compress existing scientific knowledge into models that accelerate the prediction of new hypotheses (e.g., new molecules with potential impact as drugs) and to discover new hypotheses that human scientists would not consider because of knowledge bias. As noted above with respect to the three categories of Smart Agriculture, Precision Agriculture, and Digital Agriculture, each one can take advantage of a variety of AI methods to create decision support based on captured data, including soil data, genomic and phenomics data, drone-created crop profiles, and all manner of animal management (e.g., beef, swine, fowl) [20].

To note the scope of the applications of AI to these general areas, the development of Alphafold and Foldit [21] have accelerated the identification of plausible protein folds and helped identify previously unconsidered folding hypotheses. Similarly, within the metaphor of genomics for materials [22], specific applications of current AI tools have been used to identify preferred material alloys for solar materials [23]. The same advantage arises in the identification of appropriately active small molecules in so-called "in silico" drug design [24,25].

While application domain scientists may not make these observations, the frameworks for applying AI to Digital Agriculture are identical, in the overall data science framework, to those used in protein folding, materials science, and drug design. One framework that can support this position comes from the general development of "Phenomics," arising from a 2011 workshop sponsored by the USA Department of Agriculture, and the USA National Science Foundation (NIFA, 2011). The overall framework is captured in this quote:

> "Recent advances in DNA sequencing and phenotyping technologies, in concert with analysis of large datasets have spawned 'phenomics', the use of large-scale approaches to study how genetic instructions from a single gene or the whole genome translate into the full set of phenotypic traits of an organism".

This framework provides the basis for the general study of all transformations from genome to phenome, in both animal and plant models, and creates a foundation for the crop science aspects of Digital Agriculture anticipated in this paper. At the next level of scientific granularity, AI has already been applied to understand the causality between bovine genomics and phenomics properties, and it can be similarly applied to crop

genomics [24,26]. In this regard, the promise of crop phenomics within Digital Agriculture lies within the increasing sophistication of predictive data models that approximate the causal connections between genome and phenome.

### 3.3. Synopsis of Data Driven Methods

Overall, the deployment of analytic methods to Smart Agriculture is rapidly changing, since progress in so many fields (e.g., sensing, robotics, flight-based data capture) is happening at the intersection of data collection and the basic biological science of animal and crop genomics [27,28].

Just across these two aspects of scientific endeavors, the emerging tools for analytics based on machine learning have been used in both the consolidation and prediction of crop models and in the analysis of the path from genome to phenome [29,30].

We will continue to see many new technologies linked together; this means that, increasingly, waste will be minimized, productivity will be maximized, and the impact on the environment will shrink. Success in these aspects will depend on setting standards and metrics to ensure that the progress is captured and made in a desirable direction and that technologies (current and future) will continue to drive productivity and sustainability [31].

## 4. Discussion and Perspective

### Challenges to Agricultural Digital Transition

The three categories that are needed for the adoption of SA systems include software (e.g., apps, analytic methods), hardware (e.g., connectivity device(s), sensors, wearables), and technical support services (e.g., consulting or subscription management services). For effective SA adoption, the first two are necessary, as without connectivity and data collation coupled with real-time analysis, the methods lose the "smart" aspect and revert to precision agriculture.

In addition, there are six assets across farm environments that can be connected, including (i) soil, (ii) plants, (iii) livestock, (iv) environment, (v) equipment, and (vi) people [32].

Smart agriculture ecosystems include data capture points, internet access, and data collecting/process centers to support smart agriculture. It is estimated that 50K farms require a minimal investment in their connectivity/ability to tap into the smart agriculture technology. Which is necessary to provide a foundational platform for SA.

As a historically disconnected industry that made progress based on best practices established from trial and error and industry expertise, agriculture has previously lacked the ability to make management decisions based on big data in real time. The connectivity of data from sensors, the climate, and imaging using low-power wide-area networks (LPWANs), cellular networks, and other wireless telecommunication is the backbone behind smart agriculture.

The incremental development of this foundation faces at least the following challenges:

- Dealing with legacy technology;
- Connectivity in rural areas;
- Maintenance and servicing of technologies;
- Compatibility (also as a system of systems);
- Need for standards;
- Modernizing infrastructure;
- Adoption/ability to adopt.

**Need for 5G:** Amongst the technological challenges faced across these seven challenges is the issue of nurturing the evolution of connectivity, which is currently focused on the deployment and exploitation of fifth-generation wireless communications (i.e., 5G). Some of the benefits that 5G technology can provide include the opportunity for a massive increase in bandwidth (enabling capture and movement of higher density data like images), an order-of-magnitude reduction in latency (enabling advances in real-time decision support),

and a higher capacity for multiple users (enabling more cost effective deployment over rural and remote areas).

In summary, compared to the current 4G standard, 5G will create a bigger pipe, lower latency, and faster response time. This will impact agriculture in ways that we cannot even imagine right now, particularly in terms of connectivity as it will enable real-time analytics and decision support.

**AI and Machine Learning:** The current rapid evolution of artificial intelligence and machine learning (AI&ML) has primarily promoted the development of large language models (LLMs), but the technologies for their construction and deployment are equally—and even more appropriately—valuable in application verticals like SA, where training data capture is constrained by specific challenges and contexts noted above (e.g., in the six SA assets). This means that the emerging spectrum of AI technologies, from knowledge base construction to deep learning classification, can support the foundation and development of AS.

**IoT and Ag:** As already summarized above, the incremental development of IoT data sensors and network infrastructure is a crucial component of SA foundation and development, and, like AI&ML, has direct and immediate applications within the scope of SA challenges, e.g., in the instrumentation of environment sensor networks to inform predictive systems about crop conditions.

**Low margins in agriculture:** The historical development of the economic structure of agriculture has created a complex and multi-layered economic model where the low margins between layers makes capital investment a challenge. Therefore, the pace of innovation has not kept up with other industries and today agriculture remains the least digitized of all major industries [10].

**Who owns the data?** The question of data governance and security is increasingly important as the consolidation of agriculture's economic layers means that capital investment is increasingly dependent on data capture and exploitation. For example, will large agricultural corporations increasing their grip on data lead to price discrimination? Similarly, which data owners will be most vulnerable to the increasing threat of hacking and cybersecurity disruption, which is an ever-increasing threat across all data-driven organizations.

In summary, the development of an SA foundation must address all of the above challenges simultaneously, including (1) adoption likelihood/palatability among producers; comes down to communication; (2) potential widening economic gaps that might emerge as an unintended consequence of AI adoption and other concerns, including misuse of AI and data privacy, data sovereignty; (3) the use of FinTech in agriculture to promote growth, enhance financial inclusion, and improve regional economic integration; (4) quality management for both processes and organizations; and (5) the integration of continuous improvements as those processes and organizations evolve (e.g., ISO; Six Sigma; Continuous Improvement; TQM).

## 5. The Expected Value and Development of Smart Agricultural Ecosystems

### 5.1. The Anticipated Value of Developing the Smart Agriculture Ecosystem

Climate Friendly Agriculture/Digital + Regenerative

Climate Smart Agriculture (CSA), an approach for sustainable food security, refers to the "actions needed to transform and reorient agricultural systems to effectively support development and ensure food security in a changing climate". The three main objectives of CSA include: sustainably increasing agricultural productivity, development, and incomes; adapting and building resilience to climate change (from farm to nation); and reducing and/or removing greenhouse gas emissions and/or increasing carbon sinks [33–35].

### 5.2. The Application and Impact of AL/ML in Smart Agriculture

5.2.1. Socioeconomic Benefits

It is clear from other leading adopters of AI/ML (e.g., health, law) that one of the positive aspects of their application id that they can actually create a new source of jobs

and economic opportunities for millions, globally, and sustainably produced food for the world. In addition, digital technologies provide the opportunity to change how food is produced, reduce information asymmetry, decrease transaction costs, increase equitability, and improve agriculture's environmental impact [36].

### 5.2.2. Digital Image and Deep Learning with Agriculture/Horticulture Industries

Seed purity is a critical parameter in maintaining a good germination rate and seed vigor. Traditionally, methods of identifying seed purity include the morphological inspection method, the field planting inspection method, the chemical identification method, and the electrophoresis technology inspection method; however, these techniques are time consuming, cumbersome, and require professionals; thus, they are not suitable for the mass analysis and nondestructive testing of seeds. Recently, however, hyperspectral imaging technology has been widely studied in seed purity detection, which has the characteristics of combining image and spectral information. Bi et al. (2022) developed a methodology for maize seed variety recognition based on improved swin transformer-based Deep Learning [37]. Their report showed that, in comparison with other models, the MFSwin Transformer model achieved the highest classification accuracy results. The neural network applied in the study can classify corn seeds accurately and efficiently and the image deep learning method met the high-precision classification requirements of corn seed images and has provided a reference tool for seed identification.

Cai et al. have also reported image deep learning technology that could effectively reduce the influence of environmental factors on strawberry image segmentation and provide an effective approach for the accurate operation of strawberry picking robots [38]. They improve DeepLabV3+ for strawberry image segmentation with different maturities, introducing the attention mechanism in the backbone feature extraction network and ASPP module, respectively, and adjusting the weights of feature channels in the neural network propagation process through the attention mechanism, which could enhance the feature information of strawberry images with different maturities using fewer parameters, reducing the interference of environmental factors [39].

### 6. Conclusions

As with any global challenge, integrating such a broad variety of data-driven technologies is not a simple task. But the overall anticipated value of developing a guiding framework for the creation of a SA foundation has incredible positive benefits, including:

- The role of digital technologies as a catalyst of change in how we produce food and practice agriculture/horticulture/livestock management;
- The improved efficiency of agriculture/horticulture/livestock production;
- Incorporate environmental considerations from the beginning and from all aspects;
- The ability to link all aspects of the value chain in real time.

The emerging applications of digital technology in agriculture and biology, include, but are not limited to, agriculture, data collection, data mining, bioinformatics, genomics, and phenomics, as well as applications of machine learning and artificial intelligence.

The development of a community to support this goal requires the cross linking and integration of multiple sources of agricultural research across 3S technologies (remote sensing—RS; geographic information systems—GIS; global positioning systems—GPS). This provides a basis for the detection of crop pathogens, weeds, and pests (insects) using multi-spectrum techniques and the exploitation of remote sensing technology to create and analyze multiple heterogeneously structured data sets, which enables effective cross-linking and phenomic classification. It is essential to study growth models of plants/crops and utilize expert support to develop production and smart management decision systems to achieve real-time, quantified, and precise decisions.

Topics of high interest include the capture and curation of biological "big data", research on multi-spectrum analysis, the assembly of complex genetic sequencing fragments, and structural gene predictions coupled with intermediate structures to predict phenotypes.

In this context, novel data structures are required to capture predictive structures in the path from genome type to phenotype, together with new techniques to capture and identify the regularity of biological data.

Finally, multiple-sources-based monitoring and decision making for plants, water, and nutrients in agriculture/horticulture are required, with a research focus on the utilization of remote sensing and drone sensing to compute and predict plant water usage, crop and vegetable growth and development, stress status monitoring and management, and yield and harvesting decision making.

**Author Contributions:** Conceptualization, D.T., J.Z., L.O. and R.G.; methodology, D.T., J.Z. and R.G.; validation, C.B., Y.L. and R.G.; writing—original draft preparation, J.Z. and D.T.; writing—review and editing, R.G. and J.Z.; visualization, D.T. and W.C.; supervision, Y.L. and J.Z.; project administration, L.O. and W.C.; funding acquisition, J.Z. All authors have read and agreed to the published version of the manuscript.

**Funding:** Jilin Agricultural University high level researcher grant (JLAUHLRG20102006).

**Conflicts of Interest:** The authors declare no conflicts of interest.

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
