# Peer review of "Achieving the Rewards of Smart Agriculture"

_agronomy, doi:10.3390/agronomy14030452_

Round 1

Reviewer 1 Report

Comments and Suggestions for Authors

1.      In the abstract, it’s better to present the full name of IoT (Internet of Things).

2.      In the introduction part, lack of citations to support some statements. For example, lines 52 to 56 and lines 57 to 58. Please add citations accordingly throughout the whole introduction.

3.      In Figure 1, the content is not very clear. Please use a higher-resolution image

4.      In line 93, “The distinction between “smart” agriculture and agriculture using digital technologies and precision management” is very confusing. Please define “smart agriculture” and “agriculture using digital technologies and precision management”. If these two concepts are different, which the “Smart technologies in agriculture” belong to in the next sentence?

5.      The objective is not clear. The gap and importance of understanding digital agriculture is not clear.

6.      In section 2, what is the connection between precision agriculture and smart agriculture? After the definition of precision agriculture, it seems nonsense to jump from precision agriculture to smart agriculture.

7.       What is the definition of digital agriculture? Is it the same as smart agriculture?

8.      In section 3, it is better to provide some existing examples of Agriculture 4.0 and GIS applications.

9.      In section 4, why not including the section 8 and 9? They belong to the same category.

10.  In section 4, the citation from line 183 to 189 is too long. Please reformat it.

11.  Why not show examples and applications of AI in digital agriculture?

12.  Section 5 is too general, needs more definition and examples.

13.  Many spelling typos like “Agricultureaa”

Comments on the Quality of English Language

many spelling errors 

full names and abbreviations needs match with each other

Author Response

Dear reviewer,

Thank you very much for the kind suggestions.  We revised the manuscript according to your commons.

  1. In the abstract, it’s better to present the full name of IoT (Internet of Things).

Thanks. It has been revised.

  1. In the introduction part, lack of citations to support some statements. For example, lines 52 to 56 and lines 57 to 58. Please add citations accordingly throughout the whole introduction.

Thank you for the common.

  1. In Figure 1, the content is not very clear. Please use a higher-resolution image

Thanks. We have revised.

  1. In line 93, “The distinction between “smart” agriculture and agriculture using digital technologies and precision management” is very confusing. Please define “smart agriculture” and “agriculture using digital technologies and precision management”. If these two concepts are different, which the “Smart technologies in agriculture” belong to in the next sentence? 

Thanks. We have revised this.

  1. The objective is not clear. The gap and importance of understanding digital agriculture is not clear.

Thank you for the common.

  1. In section 2, what is the connection between precision agriculture and smart agriculture? After the definition of precision agriculture, it seems nonsense to jump from precision agriculture to smart agriculture. 

Thanks, we revised precision agriculture to smart agriculture.

  1. What is the definition of digital agriculture? Is it the same as smart agriculture?

Thanks for the common.

  1. In section 3, it is better to provide some existing examples of Agriculture 4.0 and GIS applications. 

Thanks, we revised

  1. In section 4, why not including the section 8 and 9? They belong to the same category.

Thanks, we revised and merged these sections.

  1. In section 4, the citation from line 183 to 189 is too long. Please reformat it. 

Thanks, we revised this .

  1. Why not show examples and applications of AI in digital agriculture? 

Thanks for the suggestion.

  1. Section 5 is too general, needs more definition and examples.

Thanks for the suggestion.

  1. Many spelling typos like “Agricultureaa”

             Thanks, we checked and revised these.

Reviewer 2 Report

Comments and Suggestions for Authors

The article topic is actual and the interest to the readers is high.  

The abstract is mainly an introduction (especially the first part), the results are not highlighted well. 

The paper chapter structure is not logical, I suggest the following logical structure: Introduction, Methodology, Results, Discussion, and Conclusion. The actual Introduction is good, but the methodology is not well structured.

There are too many chapters, I suggest concatenating chapters 3 to 9 into 2 or 3 bigger chapters. Chapters 7, and 8 are only a few sentences, in this case, use of a subchapter is suggested. Actually, after chapter 9, there is chapter 5 Conclusion.

I suggest to revise the in text citations, because there are inconveniences (eg. 119-123 there is no ref number, 183-190 there is url, instead of ref number.) 

The conclusion is clear. 

Author Response

Reviewer 2 points

The article topic is actual and the interest to the readers is high.  

The abstract is mainly an introduction (especially the first part), the results are not highlighted well. 

Thank you for the kind commons. The abstract and first part introduction have been revised to highlight the conclusion section.

The paper chapter structure is not logical, I suggest the following logical structure: Introduction, Methodology, Results, Discussion, and Conclusion. The actual Introduction is good, but the methodology is not well structured.

Article chapters have been changed.

There are too many chapters, I suggest concatenating chapters 3 to 9 into 2 or 3 bigger chapters. Chapters 7, and 8 are only a few sentences, in this case, use of a subchapter is suggested. Actually, after chapter 9, there is chapter 5 Conclusion.

The article chapters have been simplified and revised.

I suggest to revise the in-text citations, because there are inconveniences (eg. 119-123 there is no ref number, 183-190 there is url, instead of ref number.) 

The references with DOI in the citation have been revised.

The conclusion is clear. 

Reviewer 3 Report

Comments and Suggestions for Authors

Dear authors, 

I was very happy to read this perspective paper. We have a lot of reviews and research paper, and sometime we need some good perspective, as a kind of roadmap for research. 

However I believe the paper would need some improvements as listed below:

- there is some confusion about Smart agriculture and Digital agriculture. The title initially mentions DA, but the most of the paper is focused on SA. I agree that the difference is slight: nevertheless this might generate confusion in the reader and should be corrected. 

- regarding challenges (lines 269-275), I believe the authors should a bit explain how these points have been produced: based on questionnaires? based on authors experience? please provide some explanation

-  regarding challenges (lines 269-275), I believe the authors should mention and include also the exponential growth of the "DIGITIZATION FOOTPRINT", which generates problems not only in relation to the mentioned connectivity in rural areas, but also on acceptability. Indeed having an amount of data per hectare, growing exponentially year by year, is causing resistance (and lack of big data analysis skill) among farmers, which often do not perceive data as an agronomical input

- I believe the author should better clarify what is their idea of a roadmap for the future of digital/smart agriculture, both from the point of view of farmers and of scientists

- I noticed (and I agree with this) that authors have mentioned several times the climate change: however I believe it would be important to explicitely mention also the uncertainty arising from such climate change: and I believe (and maybe also the authors agree), that uncertainty is causing variability (over space and time), and the role of digital agriculture is to support management of such uncertainty and variability. 

- probably an additional graph or figure or table (summarising some of the concepts, as also done in figure 1) would help readability of the paper. 

Comments on the Quality of English Language

English is in general fine. 

Author Response

Dear reviewer,

We are very grateful for your effort and time reviewing/communing with this manuscript. Your commons are voluble and we have went through all the points and made necessary revision. Here are our response.

I was very happy to read this perspective paper. We have a lot of reviews and research paper, and sometime we need some good perspective, as a kind of roadmap for research. 

However I believe the paper would need some improvements as listed below:

- there is some confusion about Smart agriculture and Digital agriculture. The title initially mentions DA, but the most of the paper is focused on SA. I agree that the difference is slight: nevertheless this might generate confusion in the reader and should be corrected. 

We revised the manuscript according reviewer’s suggestion.  We use “Smart agriculture” and deleted “digital agriculture”.

- regarding challenges (lines 269-275), I believe the authors should a bit explain how these points have been produced: based on questionnaires? based on authors experience? please provide some explanation.

It has been revised.

-  regarding challenges (lines 269-275), I believe the authors should mention and include also the exponential growth of the "DIGITIZATION FOOTPRINT", which generates problems not only in relation to the mentioned connectivity in rural areas, but also on acceptability. Indeed having an amount of data per hectare, growing exponentially year by year, is causing resistance (and lack of big data analysis skill) among farmers, which often do not perceive data as an agronomical input.

We thank for the common, it has been revised.

- I believe the author should better clarify what is their idea of a roadmap for the future of digital/smart agriculture, both from the point of view of farmers and of scientists

Thanks for the common.

- I noticed (and I agree with this) that authors have mentioned several times the climate change: however I believe it would be important to explicitely mention also the uncertainty arising from such climate change: and I believe (and maybe also the authors agree), that uncertainty is causing variability (over space and time), and the role of digital agriculture is to support management of such uncertainty and variability. 

Agreed, we added points accordingly.

- probably an additional graph or figure or table (summarising some of the concepts, as also done in figure 1) would help readability of the paper. 

Thank for common.

Round 2

Reviewer 1 Report

Comments and Suggestions for Authors

Could be better with more examples

Author Response

Thank you for the commons.  Our intention was a short perspective to highlight what we think about smart agriculture. 

Reviewer 3 Report

Comments and Suggestions for Authors

The authors almost did not revise the paper (or probably uploaded a wrong version of the paper).

In the response to the referee authors mentioned they had implemented revisions, but I can see only a few words corrected, one figure modified and 4-5 lines added. 

Please revise the paper based on my previous recommendations. 

However I believe the paper would need some improvements as listed below:

- regarding challenges (lines 269-275), I believe the authors should a bit explain how these points have been produced: based on questionnaires? based on authors experience? please provide some explanation

-  regarding challenges (lines 269-275), I believe the authors should mention and include also the exponential growth of the "DIGITIZATION FOOTPRINT", which generates problems not only in relation to the mentioned connectivity in rural areas, but also on acceptability. Indeed having an amount of data per hectare, growing exponentially year by year, is causing resistance (and lack of big data analysis skill) among farmers, which often do not perceive data as an agronomical input

- I believe the author should better clarify what is their idea of a roadmap for the future of digital/smart agriculture, both from the point of view of farmers and of scientists

- I noticed (and I agree with this) that authors have mentioned several times the climate change: however I believe it would be important to explicitely mention also the uncertainty arising from such climate change: and I believe (and maybe also the authors agree), that uncertainty is causing variability (over space and time), and the role of digital agriculture is to support management of such uncertainty and variability. 

- probably an additional graph or figure or table (summarising some of the concepts, as also done in figure 1) would help readability of the paper. 

Comments on the Quality of English Language

English is fine

Author Response

Dear reviewer,

We are very grateful for your effort and time reviewing/communing with this manuscript. Your commons are voluble and we have gone through the points and made necessary revision. Here is our response.

I was very happy to read this perspective paper. We have a lot of reviews and research paper, and sometime we need some good perspective, as a kind of roadmap for research. 

However I believe the paper would need some improvements as listed below:

- there is some confusion about Smart agriculture and Digital agriculture. The title initially mentions DA, but the most of the paper is focused on SA. I agree that the difference is slight: nevertheless this might generate confusion in the reader and should be corrected. 

We revised the manuscript according reviewer’s suggestion.  We use “Smart agriculture” and deleted “digital agriculture”.

- regarding challenges (lines 269-275), I believe the authors should a bit explain how these points have been produced: based on questionnaires? based on authors experience? please provide some explanation.

It has been revised.

-  regarding challenges (lines 269-275), I believe the authors should mention and include also the exponential growth of the "DIGITIZATION FOOTPRINT", which generates problems not only in relation to the mentioned connectivity in rural areas, but also on acceptability. Indeed having an amount of data per hectare, growing exponentially year by year, is causing resistance (and lack of big data analysis skill) among farmers, which often do not perceive data as an agronomical input.

We thank for the common, the 4g and 5g acceptability has been gradually implemented in regions across north America and place like Australia.  The issue raised with “DIGITIZATION FOOTPRINT” will be dealt accordingly.  

- I believe the author should better clarify what is their idea of a roadmap for the future of digital/smart agriculture, both from the point of view of farmers and of scientists

Thanks for the common.

- I noticed (and I agree with this) that authors have mentioned several times the climate change: however I believe it would be important to explicitely mention also the uncertainty arising from such climate change: and I believe (and maybe also the authors agree), that uncertainty is causing variability (over space and time), and the role of digital agriculture is to support management of such uncertainty and variability. 

Agreed, we added points accordingly.

- probably an additional graph or figure or table (summarising some of the concepts, as also done in figure 1) would help readability of the paper. 

Thank for common.